# Early Hemodynamics after Aortic Valve Replacement

**DOI:** 10.3390/medicina56120674

**Published:** 2020-12-07

**Authors:** Serik Aitaliyev, Egle Rumbinaitė, Karolina Mėlinytė-Ankudavičė, Rokas Nekrošius, Vytenis Keturakis, Rimantas Benetis

**Affiliations:** 1Department of Cardiac, Thoracic and Vascular Surgery, Hospital of Lithuanian University of Health Sciences Kauno Klinikos, Medical Academy, Lithuanian University of Health Sciences, Eivenių 2, LT-50009 Kaunas, Lithuania; rokas.nekrosius@gmail.com (R.N.); vytenis.keturakis@gmail.com (V.K.); rimantas.benetis@kaunoklinikos.lt (R.B.); 2Department of Cardiology, Medical Academy, Lithuanian University of Health Sciences, Eivenių 2, LT-50009 Kaunas, Lithuania; egle.rumbinaite@gmail.com (E.R.); kmelinyte1@gmail.com (K.M.-A.)

**Keywords:** valve hemodynamics, echocardiography, AVR, patient-prosthesis mismatch, exercise testing

## Abstract

*Background and objectives*: The aims of this study were to investigate changes in the hemodynamics associated with different types of aortic prostheses and to evaluate patient-prosthesis mismatch (PPM) at rest and after exercise. *Materials and Methods*: We retrospectively analyzed 150 patients who presented with indications for aortic valve replacement (AVR) with/without concomitant surgery from March 2019 to January 2020. The study population included 90 (60%) men and 60 (40%) women (mean age, 67.33 ± 10.22 years; range, 37–88 years). Echocardiography data such as peak and mean transprosthetic pressure gradients (Gmax, Gmean), velocity (V), effective orifice area (EOA), and indexed EOA (iEOA) were derived at rest and after exercise at baseline and before discharge. The study patients performed the six-minute walk test (6MWT) on the 5th–7th postoperative day. *Results*: Stented tissue valves showed excellent performance at rest and after exercise in comparison with mechanical valves, which showed favorable hemodynamics at rest only. At the time of discharge, moderate PPM was observed in 7/74 patients (9.5%) at rest and 5/98 (3.3%) patients after exercise. None of the patients showed severe PPM. EOA and iEOA were not significantly different between the groups. However, the stented group showed more pronounced changes in EOA and iEOA after exercise, whereas the changes in the mechanical valve group did not reach significance. *Conclusions:* In the early postoperative period, mechanical valves and stented valves showed favorable resting hemodynamics. The PPM rate measured after exercise was lower than that at rest.

## 1. Introduction

Aortic valve replacement (AVR) is one of the most common procedures in adult cardiac surgery, with approximately 45,000 implantations in one year performed in the USA alone [1,2,3,4,5]. The aim of surgical valve implantation is to provide an adequate transprosthetic gradient, which improves left ventricle (LV) hemodynamics and facilitates normalization of LV parameters [6]. Aortic valve (AV) prostheses are evaluated not only on the basis of peak and mean gradients, but also by velocity and effective orifice area (EOA). However, with manufacturer sizing systems, the variable hemodynamic responses to resting and exercise conditions in ex vivo and in vivo conditions restrict adequate hemodynamic evaluation of AV prostheses [7].

Another source of debate related to AVR is the effects of patient-prosthesis mismatch (PPM) on different types of prostheses at rest and after physical activity [8]. One study claimed that mechanical valve prostheses do not show an increase in EOA after exercise [8]. The stented tissue valve has been reported to show excellent hemodynamics, similar to stentless valve prostheses, both at rest and during exercise [9]. We hypothesized that a six-minute walk test (6MWT), rather than a bicycle ramp test, would be a suitable activity for the majority of patients in our study and utilized it for exercise testing of AV hemodynamics. The aims of this study were to investigate the changes in hemodynamics associated with different types of AV prostheses and to evaluate PPM at rest and after exercise.

## 2. Materials and Methods

### 2.1. Study Population 

Symptomatic patients with moderate or severe aortic stenosis, regurgitation, or mixed AV dysfunction referred for AVR at the Cardiac Surgery Department of the University Hospital of the Lithuanian University of Health Sciences were enrolled. Patients with concomitant procedures were enrolled as well. However, patients with previous open-heart surgery, active endocarditis, and emergent conditions were excluded from the study. Preoperative and early postoperative data, including demographic data, medical history, physical examination findings, New York Heart Association (NYHA) classification status, EuroSCORE II, The Society of Thoracic Surgeons (STS) risk score, transthoracic echocardiography findings at rest and after exercise, and intraoperative transesophageal data, were obtained for each patient. Additional information pertaining to the surgical technique, valve size, type, and manufacturer was also obtained.

We retrospectively analyzed 150 patients from March 2019 to January 2020 who showed indications for AVR with/without concomitant surgery. This study population included 90 (60%) men and 60 (40%) women with a mean age of 67.33 ± 10.22 years (range, 37–88 years). The duration of the natural course of AV disease before surgery was calculated from the first hospital admission to the operation date. AV disease symptoms may include dizziness, fainting, chest pain, irregular heartbeat, and fatigue after exercise. Preoperatively, the majority of patients were in New York Heart Association (NYHA) classes II (47.3%) and III (50.7%). Indications for AVR were as follows—stenosis (87.4%), regurgitation (11.3%), and mixed stenosis (1.3%). The tricuspid aortic valve was observed in 66% of patients and bicuspid pathology was present in 31.3% of patients. More than 90% of the patients had degenerative AV disease, whereas 2.7% had rheumatic disease.

According to prosthesis type, the patient population was divided into stented (*n* = 113), mechanical (*n* = 34), and stentless (*n* = 3) valve groups. However, due to the small number of patients in the stentless valve group, this group has been excluded from our analysis.

All human sections were acquired from the University Hospital of the Lithuanian University of Health Sciences. The Regional Medical Research Ethics Committee of the Lithuanian University of Health Sciences (No.BE-2-69, 17 September 2019) approved this research protocol. Informed consent was obtained from all patients. 

### 2.2. Surgical Techniques

All procedures were performed using a standard general anesthesia protocol. After endotracheal intubation and jugular vein and radial artery cannulations, transesophageal echocardiography was routinely used for cardiac function monitoring. All patients underwent AVR with or without concomitant surgery using standard cardiopulmonary bypass (CPB) via median (*n* = 147) or partial upper sternotomy (*n* = 3) under moderate hypothermia (*n* = 146) or hypothermic circulatory arrest (*n* = 4). Patients who required an aortic root enlargement procedure were excluded from the study. Myocardial protection was performed by selective administration of St. Thomas’ Hospital cardioplegic solution into the coronary artery or the aortic root (initial dose, 15 mL/kg; maintenance dose, 10 mL/kg). Cardioplegia was repeated every 30 min using a cross-clamp.

Cardiopulmonary bypass (CPB) under moderate hypothermia (30 °C–32 °C) or hypothermic circulatory arrest (27 °C) was performed using a roller pump (Stockert S5; Sorin Group, Munich, Germany) and oxygenator (Sorin Group, Mirandola, Italy). Heparin (3–4  mg/kg) was administered and supplemented as required to maintain an active clotting time of ≥480  s. Heparin, mannitol, and sodium bicarbonate were added to the circuit as the primary solution. CPB was performed at flow rates of 150–200  mL/kg/min, with cooling at the rate of 1 °C/min and rewarming at the rate of 1 °C every 3 min. The ascending aorta and right atrium were cannulated to establish CPB. In some cases involving AVR and concomitant procedures, two separate venous cannulas in both cava veins were utilized to provide blood return. The left atrium drained via the right upper pulmonary vein or via the pulmonary trunk. After aortic cross-clamping was applied, transverse aortotomy was performed just above the sinotubular junction, the valve was excised, and meticulous calcium debridement was performed. The prosthesis was selected after measurement using manufacturer-provided sizers. The CPB and aortic cross-clamping times were recorded.

Indications for weaning from CPB have been standardized at our institute and include stability of blood pressure, recovery of wall motion, and ST-segment normalization. A dose of protamine was used to reverse the effects of heparinization. Patients with ventricular fibrillation after aortic unclamping were electrically cardioverted with 10- to 20-joule direct shocks. A deairing procedure through the aortic root and left atrial vents was completed and checked by transesophageal echocardiography. In addition, transesophageal echocardiography was also performed in all patients to confirm the adequacy of replacement, including the presence of a paravalvular leak and prosthesis function, and assessment of ventricular function. After removal of the cannulas, myocardial pacing was positioned. Stainless steel wires (6-calibre for men and 5-calibre for women) were passed transversely through the sternum. Drains were typically left in the anterior mediastinum and pericardium, and the incision was closed.

The decision to use a tissue or mechanical valve prosthesis and the type of valve selected was based on the preference of the surgeon and patient. The majority of valves (96.7%) were implanted into the intra-annular position with continuous running 2-0 Prolene sutures, whereas 3.3% of the valves were implanted in the supra-annular position with interrupted pledgeted mattress sutures. Isolated AVR was performed in 47 (31.3%) cases. Concomitant procedures included coronary artery bypass graft (CABG) with or without concomitant surgery in 79 (52.7%) patients, whereas 24 (16%) patients required other procedures for the mitral valve (MV), tricuspid valve (TV), ascending aorta, congenital inter-septal defects, and carotid artery.

The Bio-Bentall procedure involves aortic root and ascending aorta replacement with Medtronic Freestyle stentless porcine valves or biological valve conduits with a linear ascending aortic prosthesis. The Mech-Bentall procedure refers to mechanical valve conduit implantation and ascending aorta replacement. Left and right coronary ostia were implanted into the prosthesis with 5/0 and 4/0 Prolene sutures, respectively. Mitral valve (MV) annuloplasty was standardized in our center and was performed using double semi purse-suture annuloplasty with Teflon pledget reinforcement [10].

The mean cross-clamp time and mean cardiopulmonary bypass time were 70.9 ± 26.16 min and 112.47 ± 43.04 min, respectively. Four patients underwent AVR under circulatory arrest for 12.75 ± 4.9 min (9–20 min). All patients with a mechanical prosthesis or tissue valve prosthesis with atrial fibrillation (AF) received long-term warfarin treatment with a target of an international normalized ratio (INR) of 2.0–2.5. Antithrombotic therapy with two months of warfarin administration was given in the early postoperative period to patients who received an aortic tissue valve prosthesis.

Figure 1 shows the valve types and their size distribution. Aortic valve prostheses were divided on the basis of their structure into stented tissue, stentless tissue, and mechanical prostheses. The mechanical valve prostheses included St. Jude Regent (*n* = 16, 10.7%), Sorin Carbomedics (*n* = 14, 9.3%), St. Jude Master (*n* = 2, 1.3%), Medtronic ATS (*n* = 1, 0.7%), and St. Jude Master Aortic Valved Graft (*n* = 1, 0.7%), and were implanted in 34 (22.7%) patients. The stented valve prosthesis group included St. Jude Epic (*n* = 4, 2.7%) and St. Jude Trifecta (*n* = 109, 72.7%) and were implanted in more than 75% of the study patients. The stentless aortic valve prosthesis group received the Medtronic Freestyle valve (*n* = 3, 2%). The mean prosthesis size was 24.83 ± 1.95 mm. Projected indexed effective orifice area (iEOA) data retrieved from the manufacturer provided effective orifice area (EOA) data compared to the body surface area (BSA) [11].

### 2.3. Echocardiographic Analysis 

Patients with an AV prosthesis (*n* = 150) underwent Doppler echocardiography at rest. Echocardiography data such as maximum and mean transprosthetic pressure gradients (G max, G mean), velocity (V), effective orifice area (EOA), and indexed parameter (iEOA) were derived at baseline and before discharge at rest and after exercise. Color Doppler flow images were obtained in the apical and parasternal views. Maximum and mean transprosthetic gradients were measured using the Bernoulli equation: maximum gradient (mmHg) = 4 × (V_AVmax_^2^ − V_LVOTmax_^2^) and mean gradient (mmHg) = 4 × (V_AVmean_^2^ − V_LVOTmean_^2^), where V = transaortic or transprosthetic velocity. EOA and iEOA were assessed using the continuity equation for the velocity time integral (VTI) at rest and right after exercise. EOA was calculated using the formula EOA = (CSA_LVOT_VTI_LVOT_)/VTI_AV_, where CSA is the cross-sectional area in cm^2^ and VTI is the velocity time integral [12]. The term ‘iEOA’ refers to EOA divided by BSA. PPM was considered severe when iEOA was <0.65 cm^2^/m^2^, moderate for iEOA of 0.65–0.85 cm^2^/m^2^, and absent for iEOA >0.85 cm^2^/m^2^. 

The following LV measurements and aortic valve parameters were obtained in all patients: LV end-diastolic diameter, LV septal and posterior thicknesses, LV mass, and aortic annulus. The LV ejection fraction (EF) was determined using the Simpson biplane method. Low LVEF refers to EF < 45%. To assess LV diastolic function, measurements of the E-wave, E/A ratio, and E/e were included in the study protocol. Left ventricular function was assessed from the short axis of the parasternal view, and left ventricular hypertrophy (LVH) was defined by linear measurements as 95 g/m^2^ in women and 115 g/m^2^ in men [13].

All Doppler measurements were averaged during sinus rhythm for three cardiac cycles and for five cardiac cycles with rhythm disturbance. Doppler echocardiography during the early postoperative and follow-up visits was performed with a protocol developed for this study. Transthoracic imaging was performed by one of three highly trained sonographers using the Philips EPIQ 7G and Philips CX50. For each case, 2D images and color-flow Doppler in multiple views were included.

Ninety-eight (65.3%) patients underwent exercise echocardiography with the six-minute walk test (6MWT). Doppler measurements for Gmax, Gmean, V, EOA, and iEOA were obtained in all patients within 2 min after termination of exercise.

### 2.4. Six-Minute Walk Test

Exercise tests were performed according to the American Thoracic Society guidelines [14]. Study patients performed 6MWT on the 5th–7th postoperative day. Patients were allowed to take medication and have a light meal prior to the test. Blood pressure and blood saturation were measured before and after the test. In accordance with the protocol, the patients rated shortness of breath and fatigue level on the 10-grade Borg scale before and after the test.

Subjects were instructed to walk at their own pace and to try to cover the distance as much as possible. The exercise was symptom-limited and terminated based on maximal patient effort. Since patients were allowed to stop and rest while walking the 30-m corridor, investigators encouraged subjects using standardized phrases and informed them about the remaining distance. Walking distance was used to calculate the predicted distance, as shown by Enright et al. [15]. These gender-specific equations and walking distance represent the walk percentage for healthy adults.

### 2.5. Data Analysis 

All normally distributed data were expressed as mean ± standard deviation (SD) and numbers (percentages). Continuous data without a normal distribution were presented using the median with the interquartile range (IQR). Differences between continuous variables were tested using Student’s test or the Mann–Whitney test, depending on the distribution of the data. Differences between categorical variables were evaluated using chi-squared and Fisher’s exact tests. Differences were considered significant when the P-value was less than 0.05. All statistical analyses were performed using IBM SPSS Statistics for Windows, version 26.0 (IBM Corp., Armonk, NY, USA).

## 3. Results

### 3.1. Patient Characteristics

Preoperative and operative characteristics of the study patients are shown in Table 1. The mechanical and stented tissue valve groups showed a significant sex-related difference. Stented tissue valve prostheses were more frequently used in female patients, whereas mechanical valve prostheses were predominantly used in male patients. The stented valve group represented the older patients in the study population, whereas the mechanical valve group represented the younger patients. Moreover, the prevalence of arterial hypertension (AH) and ischemic heart disease (IHD) were greater in the stented group of patients than in those who received mechanical values. Similarly, the STS score and EUROSCORE were much higher in the stented group than in the mechanical valve group. The stented tissue valve group showed a longer ICU stay than the mechanical valve group. Tricuspid aortic valve (TAV) dominated the pathology in the stented group, whereas bicuspid aortic valve (BAV) was predominantly present in the mechanical valve group. The incidence of hospital mortality was higher in the stented tissue valve group than in the other group.

Table 2 depicts the hemodynamic profiles associated with the two types of prostheses. Preoperatively, the aortic annulus and ascending aorta were significantly different between stented and mechanical valve groups. In the early postoperative period, stented and mechanical valve groups showed favorable resting hemodynamics. None of the patients showed severe PPM. EOA and iEOA were not significantly different between the groups. However, the peak and mean aortic gradients and transprosthetic velocity of the stented group were significantly different from those of the mechanical valve group. Surprisingly, the mean projected EOA of the stented group was statistically less than those of mechanical valve group. Moreover, early postoperative left ventricular function and size were similar in both groups.

### 3.2. Hemodynamic Performance after Exercise

Valve hemodynamics at rest and right after exercise in the entire study population are illustrated in Appendix A
Figure A1. Data analysis showed a significant difference between EOA and iEOA under different hemodynamic conditions, whereas V, Gmax, and Gmean were similar at rest and after exercise.

In the separate group analysis (Table 3), the stented tissue valve and mechanical valve groups demonstrated excellent hemodynamics. More pronounced changes were observed in the EOA and indexed EOA in the stented group after exercise, whereas the changes in the mechanical valve group did not reach significance.

### 3.3. 6MWT Data

Table 4 shows the distance walked at discharge and during follow-up. In our study population, 17 patients did not enter the study protocol; six patients died during hospitalization, four patients refused further study participation, and seven patients were unable to undergo the test due to postoperative stroke or a history of stroke. The endpoints of the exercise test were shortness of breath (*n* = 12), pain (knees, hips, etc., *n* = 3), and dizziness (*n* = 1). Ninety-nine (65.6%) patients were analyzed at discharge. Fifty-two patients (34.7%) were unwilling to perform the 6MWT as a result of postoperative fragility. Although the mean predicted distance was 800.75 ± 91.52 m, only 32.8% ± 12.57% of that distance was covered in the early postoperative period. There was no statistical difference between the PPM and PPM-free groups in the distance walked at discharge. 

### 3.4. PPM Rate

Moderate PPM was observed at rest in 7/74 patients (9.5%) and after exercise in 5/98 (3.3%) patients at the time of discharge. None of the patients showed severe PPM during the study. Figure 2 depicts moderate PPM in 23-mm Sorin Carbomedics and Epic St. Jude prostheses and the 25-mm St. Jude Trifecta prostheses at rest.

Figure 3 shows the detected PPM cases after physical exercise. Moderate PPM was detected in patients with 21-mm, 23-mm, and 25-mm Sorin Carbomedics prostheses and 25-mm and 27-mm St. Jude Trifecta prostheses.

## 4. Discussion

In our study we showed favorable hemodynamics of stented tissue valves compared to mechanical valves at rest. The EOA and iEOA increased after exercise in both groups, but not statistically significantly in the mechanical valves group. Although PPM was rare in our study, the PPM rate decreased after the exercise test, compare to those at rest.

We illustrated the ‘real world’ hemodynamic performance with different valves and patient profiles at rest and after exercise. We conducted a retrospective study in a large cohort of patients with AV disease from 1000 open-heart surgery centers to examine hemodynamic performance under different physical conditions. We did not restrict study participation on the basis of patient age, valve pathology, type of operation, or prosthesis selection, and we analyzed unselected data from the general Lithuanian population.

The surgical aim of AVR is to improve the long-term patient status, not to solve the aortic valve problem. For surgeons, the AVR is not an endpoint itself. The primary objectives of AVR are to decrease mortality and improve quality of life. To our knowledge, this is the first study that utilized 6MWT to compare mechanical, stented, and stentless aortic valve prostheses. Previous studies on this topic used a maximal ramp upright bicycle test [16,17,18]. Another study compared the exercise capacity of patients with AS before and after transcatheter aortic valve implantation (TAVI) by 6MWT [19].

### 4.1. Valve Hemodynamics at Rest and Exercise

Although hemodynamics at rest and after exercise were similar in our study, EOA and iEOA increased significantly after exercise. The stented tissue valve group showed better performance after exercise, whereas the mechanical valve group showed favorable hemodynamics at rest. In a seminal study by Ericiksson et al., the EOA of the mechanical valve did not change after exercise [20]. Similarly, in our study, the mechanical valve showed less pronounced changes after exercise.

Stented prostheses showed superior hemodynamics compared to the mechanical prosthesis. These prostheses have inherently lower transprosthetic gradients and velocity compared to mechanical valves. Surprisingly, despite the large EOA and iEOA of stented valves, they did not show significant differences in comparison with mechanical valves.

One of the main differences between stress echocardiography and our method to evaluate hemodynamic performance was the lack of an assessment for transprosthetic gradient changes during the test. Regardless, the 6MWT can be utilized for the hemodynamic assessment of aortic valve prostheses.

### 4.2. 6MWT Performance

Since walking is a usual exercise for the general population, especially sexa-, septua-, and octogenarians, we considered the use of this daily life activity to check prostheses performance. Thus, we utilized the 6MWT as an exercise test and reduced the possible sources of bias. To the best of our knowledge, this is the first study to use 6MWT as an exercise test to check valve hemodynamics. In our study, we did not find a difference between the distance covered in the PPM and PPM-free groups.

### 4.3. PPM Rate

Despite the 54% moderate PPM rate observed in a recent study [21], our study demonstrated a moderate PPM rate of 9.5% at rest and 3.3% after exercise. We speculate that the low PPM rate might be related to the excellent hemodynamic conditions afforded by the widely-used St Jude Trifecta valve, which was used in more than 70% of the patients. Several studies have suggested that the St Jude Trifecta valve shows favorable hemodynamics and is the best option for the aortic valve position [9,22,23]. In a systematic review and meta-analysis by Phan et al., particular attention was paid to the favorable mean gradient and EOA obtained with the Trifecta valve [24]. On the other hand, the occurrence of PPM after exercise was mostly associated with the Carbomedics mechanical valve. The main reason for this finding is that the stiff annulus cannot enlarge during exercise.

The incidence of PPM has declined over time with awareness of its negative consequences and the availability of suitable valve alternatives [21]. Thus, PPM in most cases is surgeon-controlled and can be avoided at the time of surgery. According to our data, the absolute number of PPM patients decreased after the physical exercise test. This can be explained by the hydraulic equation G = Q^2^/(k × EOA^2^), where G is the gradient, Q is the flow, and k is a constant [25]. Because of unchanged transprosthetic gradients before and immediately after exercise and the high flow (high cardiac output) after exercise, EOA increased and the PPM rate decreased. We conclude that the 6MWT can be used as an easy, available, and reliable tool to assess valve hemodynamics.

### 4.4. Limitations

We could not obtain detailed echocardiographic data for the patients who died during hospitalization. Moreover, only a limited proportion of the study population underwent 6MWT and EOA measurements after the exercise test.

## 5. Conclusions

Stented tissue valves show excellent performance at rest and after exercise in comparison with mechanical valves, which show favorable hemodynamics only at rest. The PPM rate measured after the exercise test was lower than that under resting conditions.

## Figures and Tables

**Figure 1 medicina-56-00674-f001:**
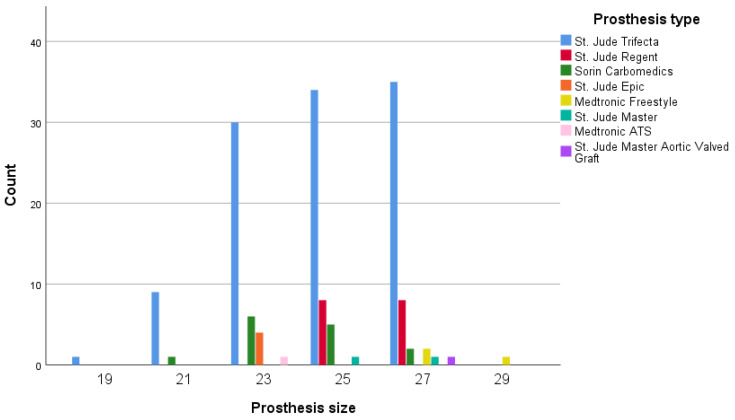
Types and sizes of prostheses.

**Figure 2 medicina-56-00674-f002:**
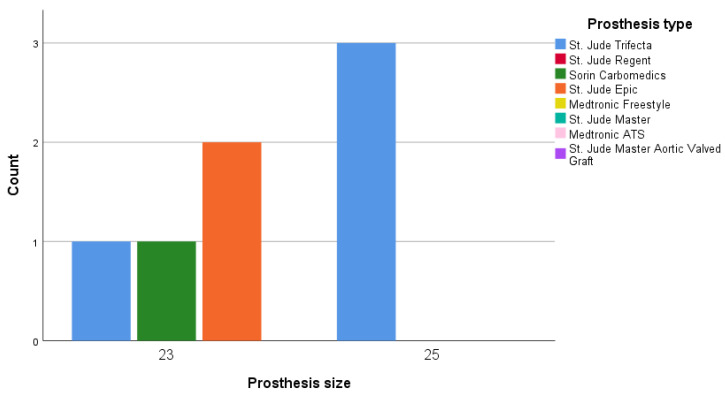
Patient-prosthesis mismatch (PPM) distribution between size and prosthesis at rest.

**Figure 3 medicina-56-00674-f003:**
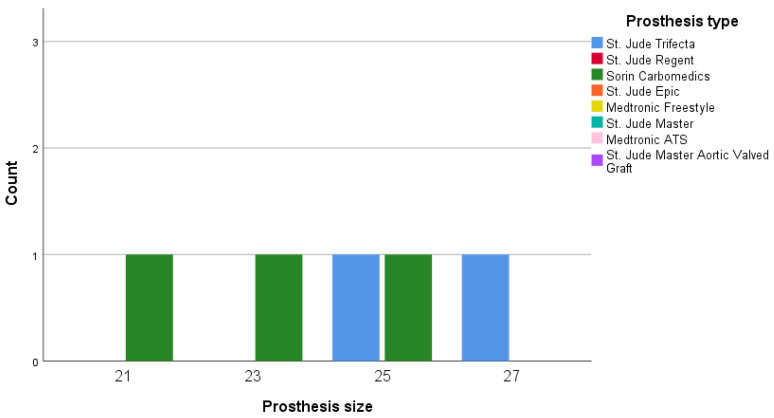
PPM distribution between size and prosthesis after 6MWT.

**Table 1 medicina-56-00674-t001:** Baseline characteristics of groups stratified by prosthesis type.

Variables		Mechanical Valve (*n* = 34)	Stented Tissue Valve (*n* = 113)	*p*-Value
Gender				
	Female	8 (23.5%)	52 (46.0%)	<0.05
	Male	26 (76.5%)	61 (54.0%)	<0.05
Age (yr)		54.47 ± 9.03	71.27 ± 6.98	<0.001
BMI (kg/m^2^)		27.14 ± 4.94	29.01 ± 5.24	NS
BMI grade (kg/m^2^)				
	Normal (18.5–25)	13 (38.2%)	23 (20.4%)	NS
	Underweight (<18.5)	No	1 (0.9%)	NS
	Overweight (25–30)	12 (35.3)	52 (46.0%)	NS
	Obese (>30)	9 (26.5%)	37 (32.7%)	NS
BSA (m^2^)		1.99 ± 0.23	1.95 ± 0.24	NS
Onset of symptoms (months)		6 (0–36)	8 (1-84)	NS
NYHA (class)				
	I	1 (2.9%)	No	NS
	II	17 (50.0%)	51 (45.1%)	NS
	III	15 (44.1%)	61 (54.0%)	NS
	IV	1 (2.9%)	1 (0.9%)	NS
STS score (%)		1.11 (0.38–11.69)	2.38 (0.52–23.50)	<0.05
EuroScore II (%)		1.9 (0.56–23.50)	3.7 (0.50–42.40)	<0.05
Dominant valvular dysfunction (*n*)				
	Stenosis	25 (75.8%)	103 (91.2%)	<0.05
	Regurgitation	7 (21.2%)	8 (7.1%)	<0.05
	Mixed dysfunction	2 (1.4%)	1 (0.7%)	NS
Native valve				
	TAV	14 (42.4%)	82 (74.5%)	<0.05
	BAV	19 (57.6%)	28 (25.5%)	<0.05
CPB time (min)		111.09 ± 51.13	112.27 ± 40.38	NS
Cross-clamp time (min)		70.79 ± 27.02	70.71 ± 26.21	NS
Operation time (min)		213.82 ± 65.41	212.55 ± 65.44	NS
ICU stay (day)		4 (1–7)	4 (1–49)	<0.05
Prolonged ventilation (more than 24 h)		1 (2.9%)	19 (17%)	NS
Prosthesis size (mm)				
	19	No	1 (0.9%)	NS
	21	1 (2.9%)	9 (8%)	NS
	23	7 (20.6%)	34 (30.1%)	NS
	25	14 (41.2%)	34 (30.1%)	NS
	27	12 (35.3%)	35 (31%)	NS
	29	No	No	NS
PPM				
	At rest	1 (5.3%)	6 (11.3%)	NS
	After exercise	3 (11.1%)	2 (2.9%)	NS
Types of operation				
	AVR	12 (35.3%)	35 (31%)	NS
	AVR + CABG ± concomitant surgery	10 (29.4%)	67 (59.3%)	NS
	AVR + concomitant surgery	12 (35.3%)	11 (9.7%)	NS
ICU drainage bleeding (mL)		400.89 ± 140.185	364.29 ± 175.36	NS
Preoperative biochemical data				
	Creatinine (μmol/L)	80 (57–936)	90 (39–236)	NS
	Hb (g/L)	137.14 ± 20.21	129.89 ± 19.08	NS
	Ht (%)	40.35 ± 5.33	38.73 ± 4.17	NS
Postoperative biochemical data				
	Creatinine (μmol/L)	81 (46–964)	93 (26–286)	NS
	Hb (g/L)	100.79 ± 12.85	96.35 ± 12.03	NS
	Ht (%)	29.74 ± 3.8	28.92 ± 3.92	NS
Comorbidities				
	AH	22 (17.9%)	98 (79.7%)	<0.05
	COPD/Asthma	2 (16.7%)	10 (83.3%)	NS
	Current smoking	4 (50%)	4 (50%)	NS
	DM	2 (10%)	17 (85%)	NS
	Hyper/dyslipidaemia	17 (18.7%)	71 (78%)	NS
	History of stroke	1 (14.3%)	6 (85.7%)	NS
	Peripheral vessel diseases	3 (10.3%)	25 (86.2%)	NS
	IHD	14 (15.1%)	77 (82.8%)	<0.05
	MI	3 (12%)	22 (88%)	NS
	Kidney/Liver diseases	5 (15.6%)	25 (78.1%)	NS
Hospital stay (days)		13 (8–67)	14 (7–113)	NS
Hospital mortality		No	6 (5.3%)	NS

Abbreviations: SD, standard deviation; BMI, body mass index; BSA, body surface area; NYHA, New York Heart Association; TAV, tricuspid aortic valve; BAV, bicuspid aortic valve; CPB, cardiopulmonary bypass; ICU, intensive care unit; PPM, patient-prosthesis mismatch; AVR, aortic valve replacement; CABG, coronary artery bypass graft; Hb, hemoglobin; Ht, haematocrit; AH, arterial hypertension; COPD, obstructive pulmonary disease; DM, diabetes mellitus; IHD, ischemic heart disease; MI, myocardial infarction; NS, non-significant; STS, Society of Thoracic Surgeons.

**Table 2 medicina-56-00674-t002:** Hemodynamic performance based on the prosthesis types.

Variables		Mechanical Valve (*n* = 34)	Stented Tissue Valve (*n* = 113)	*p*-Value
	Preoperative data			
Gmax (mmHg)		69.72 ± 30.89	76.67 ± 37.84	NS
Gmean (mmHg)		47.10 ± 15.38	46.52 ± 22.33	NS
Vmax (m/s)		4.03 ± 1.11	4.22 ± 1.12	NS
EOA (cm²)		1.07 ± 0.87	0.91 ± 0.29	NS
iEOA (cm/cm²)		0.52 ± 0.33	0.47 ± 0.15	NS
LVEDD (mm)		51.30 ± 6.49	50.13 ± 7.95	NS
iLVEDD (mm/m²)		26.45 ± 4.39	26.26 ± 3.75	NS
LV mass (g)		258.83 ± 70.46	267.78 ± 61.06	NS
iLV mass (g/m²)		129.85 ± 27.39	137.93 ± 31.11	NS
LVH		20 (76.9%)	82 (84.5%)	NS
LVEF (%)		48.68 ± 10.54	48.96 ± 10.17	NS
Low EF		6 (20.7%)	19 (20.7%)	NS
Classical LF/LG		1 (2.9%)	8 (7.1%)	NS
LV septal thickness (mm)		13.08 ± 2.37	14.28 ± 3.94	NS
LV posterior wall thickness (mm)		11.93 ± 1.95	12.10 ± 1.73	NS
Aortic annulus (mm)		24.78 ± 3.26	23.58 ± 2.38	<0.05
Sinuses Valsalva (mm)		36.71 ± 5.92	36.25 ± 4.78	NS
Proximal ascending aorta (mm)		40.22 ± 5.39	37.61 ± 4.42	<0.05
E (cm/s)		69.35 ± 22.9	77.12 ± 27.27	NS
E/A		1.02 ± 0.63	1.02 ± 0.57	NS
E/E’		13.65 ± 6.42	13.04 ± 4.81	NS
	Early postoperative data			
Gmax (mmHg)		21.32 ± 8.71	14.24 ± 6.92	<0.001
Gmean (mmHg)		12.03 ± 5.67	7.24 ± 3.77	<0.001
Vmax (m/s)		2.24 ± 0.47	1.82 ± 0.41	<0.001
EOA (cm²)		2.61 ± 0.74	2.73 ± 0.82	NS
iEOA (cm/cm²)		1.28 ± 0.35	1.39 ± 0.38	NS
Projected EOA (cm²)		1.18 ± 0.38	1.05 ± 0.16	<0.05
LVEDD (mm)		49.69 ± 6.03	48.29 ± 6.23	NS
iLVEDD (mm/m²)		25.23 ± 3.54	24.97 ± 3.08	NS
LV mass (g)		242.73 ± 62.09	239.96 ± 61.01	NS
iLV mass (g/m²)		122.15 ± 28.74	122.81 ± 27.66	NS
LVEF (%)		46.47 ± 7.64	46.76 ± 8.12	NS
LV septal thickness (mm)		12.62 ± 2.1	13.1 ± 1.86	NS
LV posterior wall thickness (mm)		12 ± 1.65	12.01 ± 1.53	NS
Aortic annulus (mm)		24.87 ± 3.38	24.63 ± 1.83	NS
Sinuses Valsalva (mm)		38.05 ± 4.23	37.06 ± 4.98	NS
Proximal ascending aorta (mm)		37.38 ± 4.31	37.54 ± 4.38	NS
E (cm/s)		84.76 ± 30.29	87.65 ± 24.33	NS
E/A		1.32 ± 0.58	1.14 ± 0.43	NS
E/E’		10.98 ± 4.08	13.52 ± 4.97	NS

Abbreviations: EOA, efficient orifice area; iEOA, indexed efficient orifice area; LVEDD, left ventricular end-diastolic diameter; iLVEDD, indexed left ventricular end-diastolic diameter; LVEF, left ventricular ejection fraction; E/A ratio, the ratio of mitral E velocity to mitral A velocity; E’, early diastolic mitral annular velocity; LV mass, left ventricular mass; iLV mass, indexed left ventricular mass; V, velocity; G, gradient; NS, non-significant.

**Table 3 medicina-56-00674-t003:** Resting and exercise differences between mechanical and stented tissue valves.

Variables	Mechanical Valve			Stented Tissue Valve		
	At Rest	After Exercise	*p*-Value	At Rest	After Exercise	*p*-Value
Gmax (mmHg)	21.32 ± 8.71	22.61 ± 11.38	0.583	14.24 ± 6.92	15.06 ± 6.55	NS
Gmean (mmHg)	12.03 ± 5.67	11.44 ± 7.05	0.871	7.24 ± 3.77	7.33 ± 3.06	NS
V (m/s)	2.24 ± 0.47	2.23 ± 0.59	0.707	1.82 ± 0.41	1.91 ± 0.39	NS
EOA (cm²)	2.61 ± 0.74	2.95 ± 0.79	0.182	2.74 ± 0.82	3.28 ± 0.92	<0.05
iEOA (cm/cm²)	1.28 ± 0.35	1.49 ± 0.39	0.181	1.39 ± 0.38	1.68 ± 0.46	<0.001

Abbreviations: EOA, efficient orifice area; iEOA, indexed efficient orifice area; V: velocity; G: gradient; NS, non-significant.

**Table 4 medicina-56-00674-t004:** 6MWT details.

Variables	PPM-Free	PPM	*p*-Value
Distance at discharge (m)	282.50 (30–684)	305 (95–410)	0.991

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
