# Peer review of "Early Hemodynamics after Aortic Valve Replacement"

_medicina, 2020, doi:10.3390/medicina56120674_

Round 1

Reviewer 1 Report

I would like to thank the authors for this paper, on investigating hemodynamic changes in association to AV prosthesis and evaluating PPM related to presence/absence of exercise. The utilisation of 6MWT I believe that adds to the literature. 

My comments are the following:

The study is quite heterogeneous in the sense that the age range is broad,

the patients have different pathology and referred to different surgeries, i.e. patients referred to AAo replacement have different hemodynamics.

Patients have different valve types i.e. BAV and TAV, which leads to different hemodynamics.

Also, the fact that the cohort of patients derives from 1000 open-heart surgery centres can introduce bias.

The differential treatment (medication-drugs) can also be a bias to this study.

Lack of blood pressure data and correlation.

In the results section, the authors refer to significant differences without referring to the trend of the change.

The stentless group (n=3) is very small to be used in any statistical comparison. Statistics should be further revised from an expert on the field. 

Different cohort number of patients noticed for the different comparisons, please make sure that the numbers are referred in the text, i.e. What is the cohort number having 6MWT data? Please clarify. 

Minor comment: "Hemodynamics instead of haemodynamics”. 

Please consider the above comments for improvement of the manuscript. To conclude with, I would like to say that the paper had a nice flow, the comparison between the 3 different types of valve prosthesis was interesting, and the Tables were quite informative. 

Yours sincerely 

Author Response

Response to Reviewer 1 Comments

Thank you for giving us the opportunity to submit a revised draft of the manuscript “Early hemodynamics after aortic valve replacement” for publication in the Medicina Journal. We appreciate the time and effort that you and the reviewers dedicated to providing feedback on our manuscript and are grateful for the insightful comments on and valuable improvements to our paper. We have incorporated most of the suggestions made by the reviewer. Those changes are highlighted within the manuscript. Please see below, in red, for a point-by-point response to the reviewers’ comments and concerns. All page numbers refer to the revised manuscript file with tracked changes.

Point 1: The study is quite heterogeneous in the sense that the age range is broad, the patients have different pathology and referred to different surgeries, i.e. patients referred to AAo replacement have different hemodynamics. Patients have different valve types i.e. BAV and TAV, which leads to different hemodynamics. Also, the fact that the cohort of patients derives from 1000 open-heart surgery centres can introduce bias.

Response 1: Thank you for pointing this out. We presented the single center AV hemodynamics in diversity of AVR patients. Moreover, our aim is to access general picture of PPM rate in heterogeneous group of AVR patients in the department of 8 surgeons’ experiences.

Point 2: The differential treatment (medication-drugs) can also be a bias to this study. Lack of blood pressure data and correlation. In the results section, the authors refer to significant differences without referring to the trend of the change.

The stentless group (n=3) is very small to be used in any statistical comparison. Statistics should be further revised from an expert on the field. 

Different cohort number of patients noticed for the different comparisons, please make sure that the numbers are referred in the text, i.e. What is the cohort number having 6MWT data? Please clarify. 

Minor comment: "Hemodynamics instead of haemodynamics”. 

Response 2: Thank you for this suggestion. Indeed, it would be interesting to explore medication therapy and look on correlations. However, this is a scope for another study. Although we didn’t made detailed analysis of blood pressure, we presented arterial hypertension data.

We agree with the reviewer’s assessment regarding the small number of patients in the stentless valve group. We excluded that group from statistical analysis.

In our study ninety-eight (65.3%) patients had 6MWT. Most of the rejected patients could not participate because of postoperative fragility.

Thank you for your consideration. I look forward to hearing from you.

Sincerely,

Serik Aitaliyev

Department of Cardiac, Thoracic and Vascular Surgery, Hospital of Lithuanian University of Health Sciences Kauno Klinikos Medical Academy

Lithuanian University of Health Sciences, Eivenių 2, LT-50009 Kaunas, Lithuania

Email: aitaliyev.serik@gmail.com  Tel: +37065391485

Reviewer 2 Report

Properly conducted study on everlasting issue of patient-prosthesis mismatch, useful to give an adjunct to recent knowledge.

Few things should be interpreted more precisely: mechanical and bioprostheses represent two different patient populations - younger with logically more frequent presence of bicuspid valves, elderly with typical senile degerative stenosis. Weak point is the inclusion of 3 (!) patients with stentless valves: neither such small number can yield any reasonable observations nor any opinion can be made without knowing the technique of implantation that may be various in this valve and definitely affects the sizing of the valve. Also these patients may have dilation aortopathy which may be the reason of relatively large sizes used ! This vague small cohort should be omitted otherwise at least any conclusions towards stentless valves throughout the manuscript should be considerably restrained.

Minor misspelling - like Benthall - Bentall correctly.

Author Response

Response to Reviewer 2 Comments

Thank you for giving us the opportunity to submit a revised draft of the manuscript “Early hemodynamics after aortic valve replacement” for publication in the Medicina Journal. We appreciate the time and effort that you and the reviewers dedicated to providing feedback on our manuscript and are grateful for the insightful comments on and valuable improvements to our paper. We have incorporated most of the suggestions made by the reviewer. Those changes are highlighted within the manuscript. Please see below, in red, for a point-by-point response to the reviewers’ comments and concerns. All page numbers refer to the revised manuscript file with tracked changes.

Point 1: Few things should be interpreted more precisely: mechanical and bioprostheses represent two different patient populations - younger with logically more frequent presence of bicuspid valves, elderly with typical senile degenerative stenosis.

Response 1: Thank you for pointing this out. We presented the single center AV hemodynamics in diversity of AVR patients. Moreover, our aim is to access general picture of PPM rate in heterogeneous group of AVR patients in the department of 8 surgeons’ experiences.

Point 2: Weak point is the inclusion of 3 (!) patients with stentless valves: neither such small number can yield any reasonable observations nor any opinion can be made without knowing the technique of implantation that may be various in this valve and definitely affects the sizing of the valve. Also these patients may have dilation aortopathy which may be the reason of relatively large sizes used ! This vague small cohort should be omitted otherwise at least any conclusions towards stentless valves throughout the manuscript should be considerably restrained.

Response 2: We agree with the reviewer’s assessment regarding the small number of patients in the stentless valve group. We excluded that group from statistical analysis.

Thank you for your consideration. I look forward to hearing from you.

Sincerely,

Serik Aitaliyev

Department of Cardiac, Thoracic and Vascular Surgery, Hospital of Lithuanian University of Health Sciences Kauno Klinikos Medical Academy

Lithuanian University of Health Sciences, Eivenių 2, LT-50009 Kaunas, Lithuania

Email: aitaliyev.serik@gmail.com  Tel: +37065391485